# From Sampling to Analysis: How to Achieve the Best Sample Throughput via Sampling Optimization and Relevant Compound Analysis Using Sum of Ranking Differences Method?

**DOI:** 10.3390/foods10112681

**Published:** 2021-11-03

**Authors:** Dalma Radványi, Magdolna Szelényi, Attila Gere, Béla Péter Molnár

**Affiliations:** 1Institute of Food Science and Technology, Hungarian University of Agriculture and Life Sciences, Villányi út 29-43, H-1118 Budapest, Hungary; Gerene.Radvanyi.Dalma@uni-mate.hu; 2Plant Protection Institute, Eötvös Loránd Research Network, Brunszvik u. 2, H-2462 Martonvásár, Hungary; szelenyim@gmail.com (M.S.); molnar.bela.peter@atk.hu (B.P.M.)

**Keywords:** adsorbent, GC–MS, lettuce, sample throughput, VOC

## Abstract

The determination of an optimal volatile sampling procedure is always a key question in analytical chemistry. In this paper, we introduce the application of a novel non-parametric statistical method, the sum of ranking differences (SRD), for the quick and efficient determination of optimal sampling procedures. Different types of adsorbents (Porapak Q, HayeSep Q, and Carbotrap) and sampling times (1, 2, 4, and 6 h) were used for volatile collections of lettuce (*Lactuca sativa*) samples. SRD identified 6 h samplings as the optimal procedure. However, 1 or 4 h sampling with HayeSep Q and 2 h sampling with Carbotrap are still efficient enough if the aim is to reduce sampling time. Based on our results, SRD provides a novel way to not only highlight an optimal sampling procedure but also decrease evaluation time.

## 1. Introduction

Volatile and semi-volatile compounds play key roles in food chemistry as primarily important cues for monitoring product quality and sensory attributes such as freshness and rotten odor [1]. There are several volatile extracting methods, but over the last few years, the stir-bar sorptive, headspace sorptive extraction, solid-phase microextraction, and dynamic headspace system sampling techniques have become the most popular. With these methods, volatiles are collected either in or above the food matrix of commonly used absorptive and adsorptive materials [2]. One of the most commonly used volatile trapping methods is the dynamic headspace system (DHS). The DHS has the advantages of a gas flow that helps to efficiently accumulate analytes in the adsorbent phase and a wide variety of ready-to-use adsorbents, e.g., activated charcoal, Porapak Q, Carbotrap, HayeSep, and Tenax. The selectivity for certain compounds depends on the adsorbent type, so one’s research question determines the type one uses [1,3].

To achieve appropriate volatile organic compound (VOC) analysis, it is necessary to optimize the sampling process. There are several possibilities for visualization that allow for the graphic evaluation of data. It has recently become common practice to visualize all reliable data (including optimization data) and thereby facilitate the understanding and enhance the usability of a dataset for practitioners. Therefore, response surface methodology has become one of the most widespread visualization tools [4,5,6]. Although response surface plots are spectacular, it is sometimes difficult to choose only one optimal condition from among various factors. Multicriteria optimization provides a solution for this problem. During multicriteria decision making, one must choose one criteria from the alternatives by considering multiple independent variables. In recent years, sum of ranking differences (SRD) has been proven to be a simple yet effective method to compare multiple subjects based on a variety of criteria [7]. SRD has been extensively used in analytical chemistry, e.g., for model comparison [8], nutritional composition comparison [9], and binary similarity metrics comparison in cheminformatics [10]. A recent comprehensive paper introduced the applicability of SRD in almost all subfields of food sciences, including the solid-phase microextraction sampling of food samples [11].

In our study, we provide a fast and reliable methodology for appropriate comparisons of different sampling processes based on the most intense plant-emitted volatile organic compounds. To fulfill our aims, *Lactuca sativa* (lettuce) was chosen as a model plant due to its volatile composition. Despite its fast growth rate and easy handling, there have been a surprisingly low number of scientific papers regarding lettuce volatiles as food. In our paper, volatiles were examined from a cut lettuce plant, lettuce oil, and lettuce seeds [12,13,14].

## 2. Materials and Methods

### 2.1. Sample

*Lactuca sativa* ‘Rivalda’ was purchased from Rijk Zwaan Budapest Ltd. (Budapest, Hungary) ‘Rivalda’ was chosen due to its good shelf-life and outstanding resistance, e.g., it is not susceptible to diseases. Seeds were planted in 1.7 L pots using potting soil in a greenhouse (natural light), with temperatures of 18–25 °C. The relative humidity was approximately 40%. To avoid soil volatiles in the headspace sample, the soil was covered with thin layers of aluminum foil after germination, only letting the lettuce plant protrude through a 2 cm diameter hole in the middle.

### 2.2. Sampling Design

Volatile collection from the whole lettuce plants was done on the 60th day after sowing. The plant was wrapped with a Nalophane NA foil tube (20 µm; Kalle Hungaria Kft., Budapest, Hungary) a day before measurements. Continuous charcoal-filtered airflow (1 L min^−1^) was pulled through the system using a vacuum pump (Thomas G 2/02 EB, Garder Denver Thomas GmbH, Fürstenfeldbruck, Germany). Volatile collection traps filled with 50 mg of Porapak Q (80–100 mesh), 50 mg of HayeSep Q (60–80 mesh), and 50 mg of Carbotrap (20–40 mesh) adsorbents (Supelco, Sigma-Aldrich, 595 North Harrison Road, Bellefonte, PA, USA) were used to collect the headspace volatiles for 1, 2, 4, or 6 h. The sampling temperature of all volatile collections was maintained at 25 ± 1 °C. Before each volatile collection, the adsorbent filters were cleaned, as described by Molnár [15]. The collected volatiles were immediately extracted with 300 µL of n-hexane into a 1.5 mL vial and kept at −18 °C until gas chromatography-mass spectrometry analysis.

### 2.3. Analytical Measurements

An Agilent 6890 gas chromatograph (GC) coupled with a 5975 C MSD mass spectrometer (MS) was used with a non-polar HP-5 UI ((5%-phenyl)-methylpolysiloxane; 30 m × 0.25 mm × 0.25 μm film; J&W, Santa Clara, CA, USA) capillary column to analyze collected volatiles. A 1 μL sample was injected into the GC injector operated in the splitless mode for 30 s, with the injector temperature set to 250 °C. The oven temperature program began at 50 °C (hold for 5 min), and it was increased to 210 °C at 5 °C min^−1^ and then to 300 °C at 20 °C min^−1^ (hold for 1 min). Helium was used as carrier gas with a constant 1.0 mL min^−1^ flow. The MS source temperature was set to 230 °C, and the quadrupole temperature was held at 150 °C. Positive electron ionization (EI+) was used with an electron energy level of 70 eV. The detector was used in scan mode between 35 and 500 *m*/*z*. The MS was tuned using perfluorotributylamine (PFTB) before measurements. Agilent Enhanced MSD ChemStation software handled the GC and MS parameters. Agilent MassHunter Workstation Qualitative Analysis B.08.00 software was used for the evaluation and comparison of the chromatograms. The Agilent NIST 2017 Mass Spectral Library was used for compound identification, and two other libraries (W9N08 and W10N11) were used to verify the identification results. Kováts indices (KIs) were calculated using the C8–C20 alkane calibration standard and the identification was also verified by comparison with KI values obtained from the NIST webbook.

### 2.4. Statistical Analysis

The sum of (absolute) ranking differences (SRD) was introduced by Héberger in 2010 [7]. The basic idea of the method is to compare methods/models to a predefined golden standard using rank numbers. The SRD algorithm consists of five main steps:A golden standard (reference, benchmark) should be defined (mean, median, minimum, maximum, or a known standard).The rank transformation of the reference column and the sampling techniques should be calculated.The absolute rank differences among each sampling technique and the reference column should be calculated.The rank differences of each sampling technique should be summed. This step results in the SRD value, which introduces the deviation or distance of a given sampling technique from the reference one.The SRD values should be normalized between 0 and 100 for easy comparability between various datasets.

The first validation of the SRD method was introduced by Héberger and Kollár-Hunek in 2011 and is called the comparison of ranks with random numbers (CRRN) [16]. The CRRN generates an SRD distribution based on the number of rows of a dataset. The problem of repeated values (ties) was solved in 2013, so SRD is capable of handling datasets where ties are present [17]. Further validation processes, such as data splitting and resampling techniques, were recently introduced the developers [18]. SRD is freely available as a Microsoft Excel macro at http://aki.ttk.mta.hu/srd/ (accessed on 11 February 2021), as an R-Shiny online application at https://attilagere.shinyapps.io/srdonline/ (accessed on 11 February 2021), and as Python implementation at https://github.com/davidbajusz/srdpy (accessed on 11 February 2021). The authors of the present study used the Microsoft Excel version.

## 3. Results

### 3.1. Sampling Optimization Using Sum of Ranking Differences

Altogether, 149 compounds were found during the evaluation of total ion chromatograms. SRD was run on all compounds (*n* = 149). Compound intensity maxima were used as the reference column for SRD. 

Figure 1 shows the results of the SRD analysis. The 6 h samplings resulted in the values most similar to the reference values, e.g., the 6 h samplings collected the highest amount of volatiles. Among these, Porapak Q and HayeSep Q were ranked first, followed by the Carbotrap adsorbent. However, to maximize daily sampling capacity, one may look for shorter sampling times. Sampling with the HayeSep Q adsorbent for 4 h (H4h) also provided acceptable results. Interestingly, the 2 h sampling using Carbotrap (C2h) adsorbent and the 1 h sampling using HayeSep Q (H1h) overtook the 4 h sampling of Porapak Q (P4h). Therefore, the H4h, C2h, and H1h volatile collection methods were the most accurate and efficient of those tested.

To answer the question regarding which adsorbent and sampling time are suitable for most volatile, semi-volatile, and less volatile compounds, we split the dataset based on the elution order (retention time). Five groups were created, each group containing 30 compounds except for the last one, which contained 29 compounds. A separate SRD was run on each group (Figure 2). Such analysis enabled us to examine the effect of volatility on the adsorbent types, as the elution order could indicate volatility willingness. In general, the more volatile a compound, the more forward it is on the total ion chromatogram (the exact order depends on the column type).

The most volatile compounds were easily captured by all adsorbent types with a 6 h sampling time (Figure 2a). All other sampling procedures followed 6 h samplings. In the case of the 4 and 2 h sampling procedures, the intensity values of the most volatile compounds may decrease on the total ion chromatogram. Semi-volatile compounds (Figure 2b,c) were also captured with the 6 h sampling time, but in this case, other sampling procedures, such P1h and H4h (Figure 2b,c) and H1h and C2h (Figure 2c), showed acceptable performance, with SRD values below 20%. P6h and H6h showed the most similar characteristics to the reference, while P2h, H2h, and C4h were ranked in the last positions.

Compounds with low volatility (Figure 2d,e) were also captured with a 6 h sampling time, but in this case, Porapak Q proved to be the best adsorbent (P6h showed the most similar characteristics to the reference). Surprisingly, H2h performed the worst; this sampling process got the last position. H4h and C2h can also be accepted, especially if one wants to shorten the sampling time. These two sampling procedures also showed good performance when all compounds were analyzed together (Figure 1). In the case of compounds with low volatility, the SRD value of P4h was around 15% (Figure 1e).

In summation, the 6 h sampling time proved to be the most suitable for all volatile compounds, among which Porapak Q was the best. C4h proved to be suitable for the most volatile compounds, while H4h, C2h, and H1h could be used for the sampling of semi- and less-volatile compounds. Using shorter sampling times enables increased sample throughput, which is an important factor in current analytical laboratories.

In addition to sampling and measurement time, analysis time (data evaluation) could also be significant. In this study, 149 compounds were analyzed on 48 total ion chromatograms. Most of the time, manual integration was also needed, especially when compounds were around the limit of detection (LOD). Figure 3 illustrates the difficulties for automatic integration algorithms.

It was complicated to find some compounds at short sampling times (1 and 2 h). When sampling time was reduced, evaluation time adequately increased. In the next step, different amounts of compounds were analyzed based on their intensity values to examine whether evaluation time could be reduced (Figure 3).

According to Figure 4, decreasing the number of evaluated compounds only had a slight effect on the SRD results. Evaluating only the first 20 most intensive compounds (highlighted by bold in Table 1) proved to be enough for reliable SRD results. However, SRD was not able to differentiate the 6 h samplings (C6h, P6h, and H6h) due to the loss of information (decreased number of compounds), though it still has enough information to differentiate sampling times. Additionally, the order of procedures did not significantly change (the tendency was preserved).

The 6 h sampling proved to be the best process for all adsorbent types; however, Figure 4d shows that Porapak Q was the best for high intensity compounds.

H4h, C2h, and H1h showed consistent performance even after reducing the number of compounds included in the analysis to 20. These findings are in accordance with those we reported earlier in the case of semi-volatiles and compounds with low volatility. H4h, H1h, and C2h are useful in the case of semi-volatiles, while H4h and C2h are useful in the case of compounds with low volatility.

An interesting finding was that by reducing the number of compounds according to their volatility, e.g., keeping only the 20 most volatile compounds, the rank of C4h decreased. This means that in cases where only the most volatile compounds are in the focus of analysis, C4h samplings might be a viable option.

### 3.2. Volatile Compositions of Lactuca sativa

We found 149 volatile compounds during the total ion chromatogram (TIC) analysis. These compounds were used for data analysis. The number of relevant lettuce compounds was decreased according to their intensity values (compounds with an integrated area higher than 0.1% of total integrated area) and their identification match factors (compounds with an identification match factor higher than 80%). Retention indices (RIs) were also used for the validation of evaluation. Delta RIs were calculated by comparing the RIs obtained from the NIST webbook to the calculated RIs. Compounds with delta RIs higher than 10% were eliminated from the compound list. Exceptions were made when a compound had a higher than 90% match factor.

Some of the reported compounds have already been discussed in other studies. α- and β-pinene, D-limonene, and β-caryophyllene were reported in lettuce oil [14]; γ-elemene and D-limonene were reported in ready-to-use lettuce [13]; and γ-elemene, β-caryophyllene, and D-limonene were found in cut lettuce [12]. Our study is unique because the whole lettuce plant was examined. Furthermore, a complete list of volatile organic compounds has been established.

## 4. Conclusions

A novel multicriteria evaluation method, SRD, was used for the first time to evaluate different sampling procedures, and it provided robust and validated results regarding the rank of the sampling procedures.

SRD identified 6 h samplings (C6h, P6h, and H6h) as the optimal procedures; however, if one wants to reduce sampling time (increase sample throughput), H4h, C2h, and H1h are also viable options.

As expected, the highest differentiation among sampling procedures was achieved when all 149 found compounds were used during SRD analysis. It was found that the ranks provided by SRD showed just slight deviations when only the first 20 most intensive compounds were used. This suggests that examining only the first 20 most intensive compounds might be enough for the determination of an optimal sampling procedure. Therefore, SRD provides a novel way to not only define the optimal sampling procedure but also decrease evaluation time.

Due to its non-parametric nature, SRD is capable of determining any sampling procedure, e.g., different volatile collection traps, stir-bar sorptive, headspace sorptive extraction, solid-phase microextraction, and dynamic headspace system techniques.

An interesting future direction is the SRD analysis of a transposed input matrix. After a proper grouping of compounds (aldehydes, ketones, terpenes, etc.), SRD ranks them based on the sampling procedures. This might help to determine which compound types (groups) are most adsorbed by the procedures.

## Figures and Tables

**Figure 1 foods-10-02681-f001:**
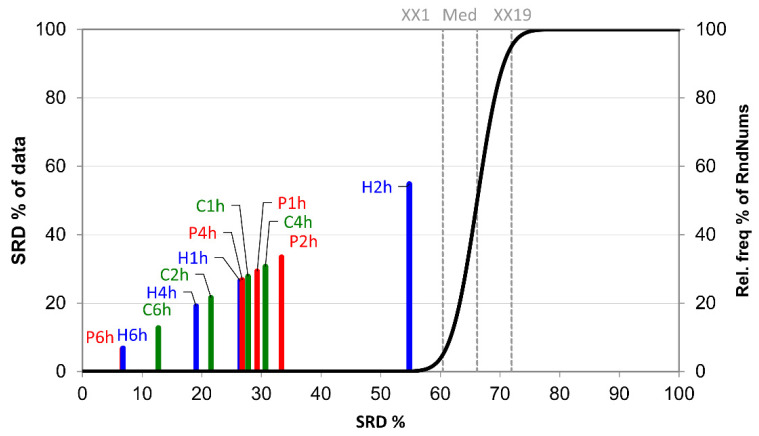
The scaled SRD values of the sampling procedure based on integrated peak area by sum of ranking differences. The maximum values of the compounds (Max) were used as the reference (benchmark) column. Scaled SRD values are plotted on the x-axis and left y-axis; the right y-axis shows the relative frequencies (black curve). Probability levels of 5% (XX1), median (Med), and 95% (XX19) are also given. If a model crosses the cumulative distribution function (XX1), say at *p* = 0.10, then the method ranks the variable as random with a 10% chance. Diagrams were produced by compound intensity values on a total ion chromatogram (*n =* 149).

**Figure 2 foods-10-02681-f002:**
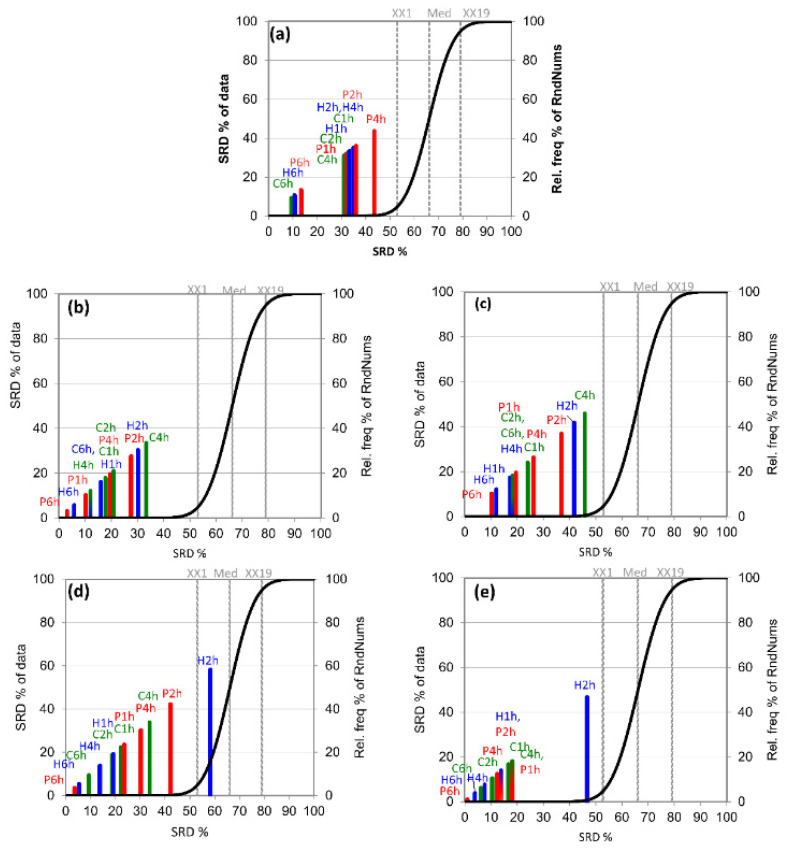
The scaled SRD values of the sampling procedure based on integrated peak area by sum of ranking differences. The maximum values of the compounds (Max) were used as the reference (benchmark) column. Scaled SRD values are plotted on the x-axis and left y-axis; the right y-axis shows the relative frequencies (black curve). Probability levels of 5% (XX1), median (Med), and 95% (XX19) are also given. If a model crosses the cumulative distribution function (XX1), say at *p* = 0.10, then the method ranks the variable as random with a 10% chance. Diagrams were produced by volatility based on the elution order of total ion chromatogram: (**a**) the first 30 compounds between 0 and 30; compounds between (**b**) 30 and 60, (**c**) 60 and 90, and (**d**) 90 and 120; and (**e**) the last 29 compounds between 120 and 149.

**Figure 3 foods-10-02681-f003:**
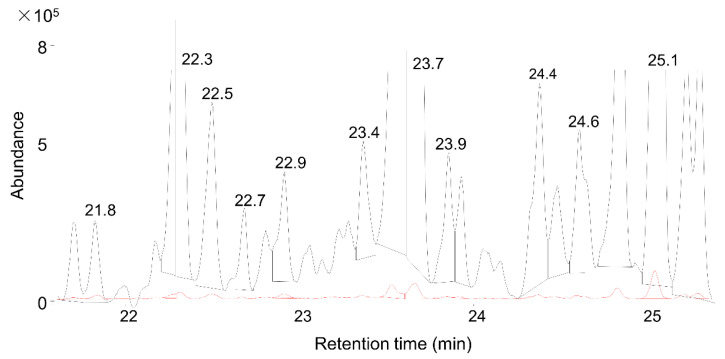
Difficulties during compound integration. The black line denotes P6h measures, and the red line denotes P1h measures. Some relevant compounds are marked with their retention times.

**Figure 4 foods-10-02681-f004:**
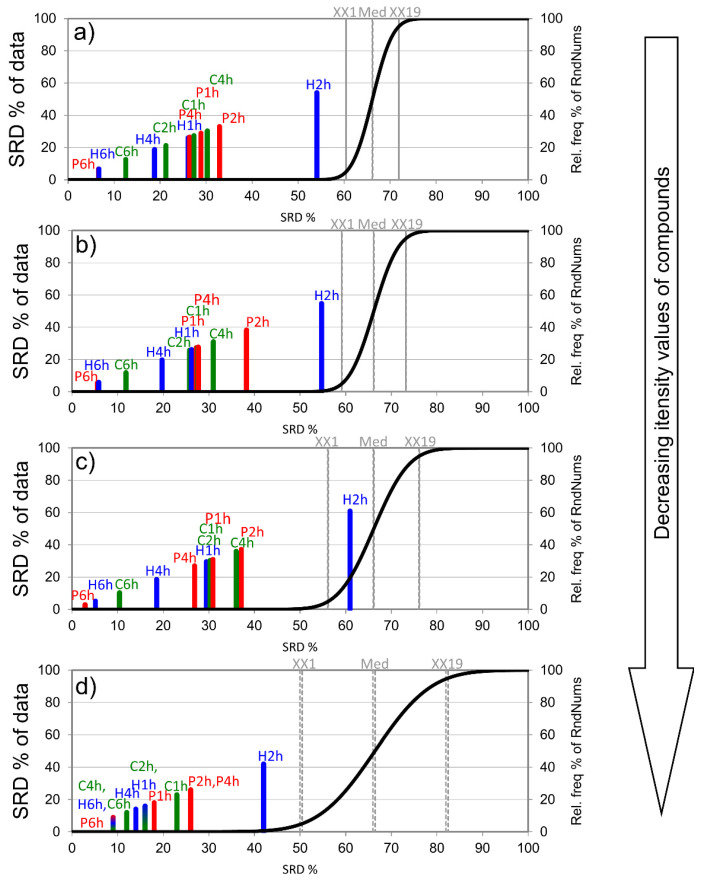
The scaled SRD values of the sampling procedure based on integrated peak area by sum of ranking differences. The maximum values of the compounds (Max) were used as the reference (benchmark) column. Scaled SRD values are plotted on the x-axis and left y-axis; the right y-axis shows the relative frequencies (black curve). Probability levels of 5% (XX1), median (Med), and 95% (XX19) are also given. If a model crosses the cumulative distribution function (XX1), say at *p* = 0.10, then the method ranks the variable as random with a 10% chance. Diagrams were produced according to intensity values: (**a**) all compounds (149), (**b**) the first 100 highest intensity compounds, (**c**) the first 50 highest intensity compounds, and (**d**) the first 20 most intensive compounds.

**Table 1 foods-10-02681-t001:** Identified volatile organic compounds in *Lactuca sativa*. Retention indices written in italics come from the NIST webbook. The 20 most intensive compounds are marked in bold. nd indicates no data.

RT (Min)	RI (Calculated)	RI (Literature)	Compound Name	Formula	CAS Number	Area (%)	Match Factor (%)
3.77	*775*	*775*	3-Hexanone	C_6_H_12_O	589-38-8	0.04	>90%
3.83	*791*	*791*	2-Hexanone	C_6_H_12_O	591-78-6	0.06	>85%
3.91	*797*	*797*	3-Hexanol	C_6_H_14_O	623-37-0	0.04	>85%
3.99	803	*800*	Octane	C_8_H_18_	111-65-9	0.15	~90%
4.12	810	*805*	1,3-Dimethylcyclohexane	C_8_H_16_	2207-03-6	0.03	>85%
4.68	838	*840*	Cyclogeraniolane	C_9_H_18_	3073-66-3	0.03	>85%
4.73	841	*842*	2,4-Dimethyl-1-heptene	C_9_H_18_	19549-87-2	0.07	~90%
5.16	863	*860*	Ethylbenzene	C_8_H_10_	100-41-4	0.28	>90%
5.32	871	*879*	o-Xylene	C_8_H_10_	95-47-6	0.56	>95%
5.78	894	*898*	Styrene	C_8_H_8_	100-42-5	0.17	>90%
5.84	897	*885*	m-Xylene	C_8_H_10_	108-38-3	0.21	>95%
5.95	902	*900*	Nonane	C_9_H_20_	111-84-2	0.25	>95%
6.45	922	*949*	1,3-Dimethyl-2-(1-methylethylidene)cyclopentane	C_10_H_18_	61142-31-2	0.12	>80%
6.76	934	*936*	2,6-Dimethyloctane	C_10_H_22_	3051-30-1	0.26	~95%
6.81	936	*937*	α-Pinene	C_10_H_16_	80-56-8	0.64	>95%
6.95	942	*941*	2-Methylheptane-3-ethyl	C_10_H_22_	14676-29-0	0.54	>80%
7.18	951	*961*	β-Pinene	C_10_H_16_	127-91-3	0.03	~75%
7.30	956	*950*	Isocumene	C_9_H_12_	103-65-1	0.30	~80%
7.50	963	*963*	p-Ethyltoluene	C_9_H_12_	622-96-8	0.25	>95%
7.53	965	*970*	2-Methylnonane	C_10_H_22_	871-83-0	0.69	>90%
7.68	970	*975*	Mesitylene	C_9_H_12_	108-67-8	0.38	~95%
7.71	972	*972*	3-Methylnonane	C_10_H_22_	5911-04-6	0.42	>95%
7.79	975	*979*	trans-p-Menthane	C_10_H_20_	1678-82-6	0.29	>90%
7.98	982	*979*	o-Ethyltoluene	C_9_H_12_	611-14-3	0.10	>90%
8.13	988	*989*	cis-p-Menthane	C_10_H_20_	6069-98-3	0.37	>85%
8.21	992	*997*	Octahydro-1H-indene	C_9_H_16_	4551-51-3	0.97	>95%
**8.34**	**997**	nd	**Benzene, 1,2,3-trimethyl-**	**C_9_H_12_**	**526-73-8**	**1.39**	**>95%**
**8.48**	**1002**	*1000*	**Decane**	**C_10_H_22_**	**124-18-5**	**3.47**	**>95%**
8.94	1019	*nd*	(±) Menthol	C_10_H_20_O	15356-70-4	0.39	>85%
**9.12**	**1025**	*1051*	**4-Methyldecane**	**C_11_H_24_**	**2847-72-5**	**2.04**	**>85%**
9.19	1027	*1030*	2-Cyclohexylbutane	C_10_H_20_	7058-01-7	0.36	>85%
9.31	1032	*1028*	D-Limonene	C_10_H_16_	5989-27-5	0.75	>95%
**9.77**	**1048**	*nd*	**1,2-Dimethylcyclooctene**	**C_10_H_18_**	**54299-96-6**	**0.86**	**>85%**
9.85	1051	*1051*	cis-β-Ocimene	C_10_H_16_	3338-55-4	0.12	>95%
10.04	1058	*1055*	Naphthan	C_10_H_18_	91-17-8	0.63	>90%
10.09	1060	*1059*	2,5-Dimethylnonane	C_11_H_24_	17302-27-1	0.83	>80%
**10.23**	**1064**	*1078*	**4,7-Methanoindan, hexahydro-**	**C_10_H_16_**	**6004-38-2**	**3.48**	**>95%**
**10.75**	**1083**	*1081*	**1,1′-Bicyclopentyl**	**C_10_H_18_**	**1636-39-1**	**1.76**	**>95%**
**11.07**	**1095**	*1083*	**3-tert-Butyltoluene**	**C_11_H_16_**	**1075-38-3**	**0.53**	**>80%**
**11.14**	**1097**	*nd*	**Unknown1 (135 *m*/*z*)**			**1.12**	
11.30	1103	*1100*	Undecane	C_11_H_24_	1120-21-4	3.22	**>90%**
12.29	1138	*1146*	2-Methyldecalin	C_11_H_20_	2958-76-1	0.72	>90%
12.75	1154	*nd*	Tricyclo[5.2.1.0(2,6)]decane, 4-methyl-	C_11_H_18_	2000073-34-9	0.70	>90%
**13.39**	**1176**	*nd*	**Toluene, p-(1-ethylpropyl)-**	**C_12_H_18_**	**22975-58-2**	**0.32**	**>85%**
13.65	1186	*1178*	Benzene, 1-methyl-2-(1-ethylpropyl)-	C_12_H_18_	54410-74.1	1.03	>85%
**14.15**	**1203**	*nd*	**Benzene, 1,4-dimethyl-2-(2-methylpropyl)-**	**C_12_H_18_**	**55669-88-0**	**1.84**	**>85%**
**17.51**	**1326**	*1304*	**2,7-Dimethyltetralin**	**C_12_H_16_**	**13065-07-1**	**1.73**	**>85%**
**17.66**	**1332**	*1348*	**6-Ethyltetralin**	**C_12_H_16_**	**22531-20-0**	**2.10**	**>85%**
**17.87**	**1339**	*1354*	**5-Ethyltetralin**	**C_12_H_16_**	**42775-75-7**	**1.64**	**>85%**
20.09	1426	*1423*	β-Caryophyllene	C_15_H_24_	87-44-5	0.25	~90%
20.32	1435	*nd*	(-)-Isolongifolol, methyl ether	C_16_H_28_O	999281-62-4	0.24	>85%
20.52	1443	*nd*	Corymbolone	C_15_H_24_O_2_	97094-19-4	0.16	>85%
21.14	1467	*1460*	α-Humulene	C_15_H_24_	6753-98-6	0.26	~80%
21.71	1488	*1465*	γ-Elemene	C_15_H_24_	3242-08-8	0.44	>75%
**22.50**	**1525**	*1456*	**β-Humulene**	**C_15_H_24_**	**116-04-1**	**1.26**	**>80%**
**23.55**	**1568**	*nd*	**Unknown2 (135 *m*/*z*)**	**-**	**-**	**4.43**	**-**
**23.68**	**1573**	*nd*	**Longifolene-I2**	**C_15_H_24_**	**1000162-76-7**	**4.88**	**~90%**
**24.38**	**1606**	*nd*	**7-Octylidenebicyclo[4.1.0]heptane**	**C_15_H_26_**	**82253-11-0**	**1.42**	**>85%**
**25.08**	**1636**	*nd*	**1,4-Methanobenzocyclodecene, 1,2,3,4,4a,5,8,9,12,12a-decahydro-**	**C_15_H_22_**	**74708-73-9**	**8.23**	**~90%**
**25.30**	**1645**	*nd*	**Cyclobuta[1,2:3,4]dicyclooctene, 1,2,5,6,6a,6b,7,8,11,12,12a,12b-dodecahydro-, (6a.α.,6b.α.,12a.β.,12b.β.)-**	**C_16_H_24_**	**61233-68-9**	**1.48**	**>85%**

## Data Availability

The data presented in this study are available on request from the corresponding author.

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
