# Peer review of "From Sampling to Analysis: How to Achieve the Best Sample Throughput via Sampling Optimization and Relevant Compound Analysis Using Sum of Ranking Differences Method?"

_foods, 2021, doi:10.3390/foods10112681_

Round 1

Reviewer 1 Report

Paper describes interesting approach in volatile's analysis of Lactuca sativa. Some remarks that should be taken into consideration:

  1. Use the same convention for units, I mean l or L;
  2. For better signal's separation authors could decrease the temp. ramp, instead of 5 to 2-3  oC at the beginning;
  3. I found several misinterpretations, e.g
  1. a) 2,2-Dimethylhexane could be solvent or other impurity; please re-check blanks;
  2. b) Methylcyclohexane could be solvent or other impurity; please re-check blanks;
  3. c) 2,5-Dimethylhexane could be solvent or other impurity; please re-check blanks;
  4. d) 2, ,4-Dimethylhexane could be solvent or other impurity; please re-check blanks;
  5. e) Ethylcyclopentane could be solvent or other impurity; please re-check blanks;
  6. f) 2,3,3-Trimethylpentane could be solvent or other impurity; please re-check blanks;
  7. g) 2,3-Dimethylhexane could be solvent or other impurity; please re-check blanks;
  8. h) 2-Methylheptane could be solvent or other impurity; please re-check blanks;;
  9. i) Toluene could be solvent or other impurity; please re-check blanks;
  10. j) 3-Methylheptane could be solvent or other impurity; please re-check blanks;
  11. k) 1,3(trans)-dimethylcyclohexane could be solvent or other impurity; please re-check blanks;
  12. l) 2,4-Dimethylheptane could be solvent or other impurity; please re-check blanks;
  13. m) Ethylcyclohexane could be solvent or other impurity; please re-check blanks;
  14. n) 2,4-Dimethyl-1-heptene could be solvent or other impurity; please re-check blanks;
  15. o) 1,2,4-Trimethylcyclohexane could be solvent or other impurity; please re-check blanks;
  16. p) 1,1,2-Trimethylcyclohexane could be solvent or other impurity; please re-check blanks;
  17. q) 1-Ethyl-4-methylcyclohexane could be solvent or other impurity; please re-check blanks;
  18. r) 1-Ethyl-4-methylcyclohexane could be solvent or other impurity; please re-check blanks;
  19. s) 1-Ethyl-3-methylcyclohexane could be solvent or other impurity; please re-check blanks;
  20. t) 1-Methyl-cis-4-ethylcyclohexane could be solvent or other impurity; please re-check blanks;
  21. u) Propylcyclohexane could be solvent or other impurity; please re-check blanks;
  22. v) 2,6-Dimethyloctane could be solvent or other impurity; please re-check blanks;
  23. w) p-Ethyltoluene could be solvent or other impurity; please re-check blanks;
  24. x) cis-Octahydro-1H-indene could be solvent or other impurity; please re-check blanks;
  25. y) Butylcyclohexane could be solvent or other impurity; please re-check blanks;
  26. z) 1,1'-Bicyclopentyl could be solvent or other impurity; please re-check blanks;
  27. aa) 3-tert-Butyltoluene could be solvent or other impurity; please re-check blanks;
  28. bb) cis-3-Methyl-endo[1]tricyclo[5.2.1.0(2.6)]decane could be solvent or other impurity; please re-check blanks;
  29. cc) cis-4-Methyl-exo[1]tricyclo[5.2.1.0(2.6)]decane could be solvent or other impurity; please re-check blanks;
  30. dd) Toluene, p-(1-ethylpropyl)- could be solvent or other impurity; please re-check blanks;
  31. ee) 2-Methyldecalin could be solvent or other impurity; please re-check blanks;
  32. ff) 6,6-Dimethyl-cyclooct-4-enone could be solvent or other impurity; please re-check blanks;
  33. gg) 5 as well as 6,6-Dimethyl-cyclooct-4-enone could be solvent or other impurity; please re-check blanks;
  34. hh) 7-Octylidenebicyclo[4.1.0]heptane could be solvent or other impurity; please re-check blanks;
  35. ii) 1,4-Methanobenzocyclodecene could be solvent or other impurity; please re-check blanks;

Author Response

  1. Use the same convention for units, I mean l or L;

Thank you for your advice, we corrected the manuscript and use the same unit L

  1. For better signal’s separation authors could decrease the temp. ramp, instead of 5 to 2-3  oC at the beginning;

Thank you for your comment, we will consider it for our next experiment. In this work, we focused mostly on the SRD statistics. The main compounds, which were interesting in this study, could be deconvoluted from the TIC chromatogram without any compound identification problems.

  1. I found several misinterpretations, e.g ...

We re-checked the blank samples and corrected table 1. Some of the compounds were corrected or removed from the list (as the final 23 compounds were excluded from table1.).

  1. a) 2,2-Dimethylhexane could be solvent or other impurity; please re-check blanks;
  2. b) Methylcyclohexane could be solvent or other impurity; please re-check blanks;
  3. c) 2,5-Dimethylhexane could be solvent or other impurity; please re-check blanks;
  4. d) 2, ,4-Dimethylhexane could be solvent or other impurity; please re-check blanks;
  5. e) Ethylcyclopentane could be solvent or other impurity; please re-check blanks;
  6. f) 2,3,3-Trimethylpentane could be solvent or other impurity; please re-check blanks;
  7. g) 2,3-Dimethylhexane could be solvent or other impurity; please re-check blanks;
  8. h) 2-Methylheptane could be solvent or other impurity; please re-check blanks;;
  9. i) Toluene could be solvent or other impurity; please re-check blanks;

10 j) 3-Methylheptane could be solvent or other impurity; please re-check blanks;

  1. l) 2,4-Dimethylheptane could be solvent or other impurity; please re-check blanks;
  2. m) Ethylcyclohexane could be solvent or other impurity; please re-check blanks;
  3. o) 1,2,4-Trimethylcyclohexane could be solvent or other impurity; please re-check blanks;
  4. p) 1,1,2-Trimethylcyclohexane could be solvent or other impurity; please re-check blanks;
  5. q) 1-Ethyl-4-methylcyclohexane could be solvent or other impurity; please re-check blanks;
  6. r) 1-Ethyl-4-methylcyclohexane could be solvent or other impurity; please re-check blanks;
  7. s) 1-Ethyl-3-methylcyclohexane could be solvent or other impurity; please re-check blanks;
  8. t) 1-Methyl-cis-4-ethylcyclohexane could be solvent or other impurity; please re-check blanks;
  9. u) Propylcyclohexane could be solvent or other impurity; please re-check blanks;
  10. y) Butylcyclohexane could be solvent or other impurity; please re-check blanks;
  11. bb) cis-3-Methyl-endo[1]tricyclo[5.2.1.0(2.6)]decane could be solvent or other impurity; please re-check blanks;
  12. cc) cis-4-Methyl-exo[1]tricyclo[5.2.1.0(2.6)]decane could be solvent or other impurity; please re-check blanks;
  13. ff) 6,6-Dimethyl-cyclooct-4-enone could be solvent or other impurity; please re-check blanks;

+1, 1,2,3-Trimethylcyclohexane

We sincerely thank your detailed work.

Reviewer 2 Report

Comments to Authors

The work is good to read and the structure of the work is clear. Generally, I don't have too many comments about the work because it is complete. A few comments You can find below.

- Lines 54-55: "To fulfill our aims, Lactuca sativa (lettuce) was chosen as a model plant due to its volatile composition". - I don't quite understand this statement. Please explain.

- Lines: 83-84. Why did the authors choose this kind of column?

- Lines: 132-133: “6-hour samplings collect the highest amount of volatiles” and lines: 189-190 “To sum up, 6-hour sampling time proved to be the most suitable for all volatile compounds, among them Porapak Q should be suggested” and lines: 228-229 “6-hours sampling proved to be the best sampling process regarding all adsorbent 228 types,…” this is an obvious and expected result. Please comment.

- Lines: 173-174. “All other sampling procedures followed 6-hour samplings, while the 4-hour sampling with Porapak Q adsorbent performed the worst”. Please explain it.

- The literature review is somewhat sparse, although most of the citations are new.

Summing up, the work is consistently implemented, the conclusions correspond to the aim of the work.

Author Response

The work is good to read and the structure of the work is clear. Generally, I don't have too many comments about the work because it is complete. A few comments You can find below.

- Lines 54-55: "To fulfill our aims, Lactuca sativa (lettuce) was chosen as a model plant due to its volatile composition". - I don't quite understand this statement. Please explain.

Our main aim was to provide a fast and reliable methodology, which is able for an appropriate comparison of different sampling processes. The differentiation was done using the most intense emitted plant volatiles. For this experiment, we have chosen Lactuca sativa as a model plant, but it could be other plants, too. The main point of this article is the methodology, not the plant itself; therefore we called it a model plant.

- Lines: 83-84. Why did the authors choose this kind of column?

We used a non-polar HP-5 UI MS ((5 %-phenyl)-methylpolysiloxane; 30 m x 0.25 mm x 0.25 μm film, J&W) capillary column to analyze the collected volatiles. This column has very

low bleed characteristics and excellent inertness for active compounds, including basic compounds. It has improved signal-to-noise ratio for better sensitivity and mass spectral integrity. Since we haven’t known the examined compounds in advance, we designed an untargeted analysis, therefore we used a general-purpose column.

- Lines: 132-133: “6-hour samplings collect the highest amount of volatiles” and lines: 189-190 “To sum up, 6-hour sampling time proved to be the most suitable for all volatile compounds, among them Porapak Q should be suggested” and lines: 228-229 “6-hours sampling proved to be the best sampling process regarding all adsorbent 228 types,…” this is an obvious and expected result. Please comment.

From our previous experiments (not published yet), we found that 4-hour sampling time was better for volatile compounds because the very volatile compounds were sucked through the filled adsorbent, and the amount of these compounds could not detect properly. However, in this experiment, we did not find this phenomenon (we used different adsorbents and slower air flow to avoid this phenomenon).

- Lines: 173-174. “All other sampling procedures followed 6-hour samplings, while the 4-hour sampling with Porapak Q adsorbent performed the worst”. Please explain it.

In our previous works, we often used 4-h sampling with Porapak adsorbent. In this project, the sampling process was optimized using a new approach. For us, it was very interesting information that for some components (e.g. the most volatile compounds) we found a better sampling option with the SRD method. This paragraph has been slightly modified as follows:

“The most volatile compounds were easily captured by all adsorbent types with a 6-hour sampling time (Fig. 2a). All other sampling procedures followed 6-hour samplings. In the case of 4-, and 2-hour sampling procedures, the intensity values of the most volatile compounds may decrease on the total ion chromatogram.”

- The literature review is somewhat sparse, although most of the citations are new.

Thank you for your comment.

Summing up, the work is consistently implemented, the conclusions correspond to the aim of the work.

We sincerely thank your detailed work.

Reviewer 3 Report

The manuscript entitled “From Sampling to Analysis: How to achieve the best sample throughput via sampling optimization and relevant compound analysis using sum of ranking differences method?” reports a fast and reliable methodology, for appropriate comparison of different sampling processes based on the most intense emitted plant volatile organic compounds from Lactuca sativa (lettuce).

Table 1. Please add the RI from literature for equivalent columns. Considering the capilar column used in this work, how can you identify a cis or a trans? Benzene, 1-methyl-2-(1-ethylpropyl)- should be 1-methyl-2-(1-ethylpropyl)-benzene, β-Caryophyllen should be β-Caryophyllene, and soon.

Author Response

The manuscript entitled “From Sampling to Analysis: How to achieve the best sample throughput via sampling optimization and relevant compound analysis using sum of ranking differences method?” reports a fast and reliable methodology, for appropriate comparison of different sampling processes based on the most intense emitted plant volatile organic compounds from Lactuca sativa (lettuce).

Table 1. Please add the RI from literature for equivalent columns.

We added literature RI values using NIST Webbook to Table 1. We used literature data from HP1, HP5, and HP 5ms (because they have the same sorbent phase) column, therefore some differences may occur between calculated and literature RI data.

Considering the capilar column used in this work, how can you identify a cis or a trans?

Thank you for your comments. This type of column cannot really separate cis/trans configuration. We used NIST library identification results in Table 1. We deleted the geometric isomerism from the list. In the case of cis/trans-p-Menthane we used cis/trans configuration because they separated, and according to their RI values, trans-p-Menthane eluated earlier.

Benzene, 1-methyl-2-(1-ethylpropyl)- should be 1-methyl-2-(1-ethylpropyl)-benzene, β-Caryophyllen should be β-Caryophyllene, and soon.

For ease of retrieval, a uniform nomenclature was used, based on NIST MS library. Table 1 is also slightly modified (some compounds were removed from the list) and we re-checked the nomenclature and some compounds' name were modified according to NIST Webbook “other names” function. In the case of β-Caryophyllen and β-Caryophyllene, which was a misspelling, we corrected in the manuscript.

We sincerely thank your detailed work.
